# Real-Life Value of the Odysight^®^ Application in At-Home Screening for Exudative Recurrence of Macular Edema

**DOI:** 10.3390/jcm11175010

**Published:** 2022-08-26

**Authors:** Gauthier Kielwasser, Laurent Kodjikian, Corinne Dot, Carole Burillon, Philippe Denis, Thibaud Mathis

**Affiliations:** 1Service d’Ophtalmologie, Hôpital Universitaire de la Croix-Rousse, Hospices Civils de Lyon, 69004 Lyon, France; 2Service d’Ophtalmologie, Hôpital Edouard Herriot, Hospices Civils de Lyon, 69003 Lyon, France; 3Laboratoire UMR-CNRS 5510 Matéis, 69100 Villeurbanne, France; 4Service d’Ophtalmologie, Hôpital D’instruction des Armées Desgenettes, 69003 Lyon, France

**Keywords:** age-related macular degeneration, intravitreal injection, macular edema, Odysight^®^, telemedicine

## Abstract

Background: The aim of this study was to describe the value of the Odysight^®^ application in addition to a classical follow-up regimen in the detection of exudative recurrences in patients with macular edema. Methods: We conducted an observational, multicenter, retrospective study. The Odysight^®^ application includes a visual acuity (VA) test that can lead to alerts in case of a drop of >5 VA letters on two successive tests. The efficacy of the alerts in detecting exudative recurrence was studied. Results: A total of 149 eyes of 123 patients were included. The sensitivity of alerts for the detection of recurrence was 30.8% (95% CI [17.6; 44.0]) and its specificity was 83.7% (95% CI [73.2; 94.3]. A better baseline VA was found to be significantly associated with a better retention of the application (OR = 0.05, 95% CI [0.002; 0.62]; *p* = 0.045). Of the 12/39 alerts that detected a recurrence, eight (20.5% of all alerts) resulted in the scheduling or advancement of an intravitreal injection. Conclusions: In the present study, the sensitivity and positive predictive value of the Odysight^®^ application seems low but its use in addition to the usual follow-up of the patient can detect, in certain cases, an early recurrence and thus allow an anticipated readjustment of the treatment.

## 1. Introduction

The prognosis of macular edema from chronic retinal diseases has considerably changed since intravitreal injections have become available [1]. Nevertheless, these diseases are treated with repeated injections to avoid recurrences that lead to a short-term visual loss which can be definitive if not treated soon enough [2,3]. Recurrence can be confirmed by visual acuity (VA) measurements, but more importantly, by optical coherence tomography (OCT), a highly specialized examination that reveals edema in the retina in a quasi-histological manner. The first follow-up regimens were based on monthly injections, and these have been replaced more recently by regimens developed to limit the therapeutic burden, such as the reactive and the proactive regimens [4]. However, recurrence can occur between two consultations for both regimens and lead to visual loss if not treated in time [5]. Tools to detect recurrence at home have been developed; the Amsler grid is the most well-known but its sensitivity to detect recurrence is quite variable and rather low for a screening test [6]. Moreover, there may be limited value in its use alone for the detection of new disease activity [7], or recurrence [8], in neovascular age-related macular degeneration (AMD). More recently, several remote tools have been developed to detect early-stage disease or recurrence. The ForeseeHome^®^ is a home device that use preferential hyperacuity perimetry, providing rather good performance to detect early presence of neovascular AMD [9]. However, it requires specific equipment that is not commercialized in Europe. Since the advance of smartphone technology, specific applications for eye monitoring have been developed. Two mobile applications using hyperacuity measure using a dot alignment test (Alleye^®^; Oculocare medical Inc., Zürich, Switzerland) or shape discrimination (myVisionTrack^®^; Vital Art and Science, Inc., Dallas, TX, USA) have been developed [10,11] but were not available in France at the beginning of the study. Odysight^®^ is another mobile application (Tilak Healthcare, Paris, France) that was first released in 2018 and which allows the measurement of VA at home on a smartphone, sending the prescriber and the patient an alert when it is found to be reduced with respect to baseline on two successive tests [12]. This could be indicative of potential recurrence and therefore bring forward the following appointment and thus allow more rapid treatment in case of confirmed recurrence. Although this tool has been evaluated for its effectiveness in detecting a drop in VA [13], no study has been conducted under real-life conditions for detecting recurrence. Moreover, it is also reported that the use of the application decreases over time (51% of users at 3 months) [13], and it is therefore of interest to identify the factors associated with retention in order to better target this application.

The aim of the present observational study was to describe the value of the Odysight^®^ application in addition to a classical follow-up regimen for the detection of exudative recurrences in patients with macular edema treated by intravitreal injection. The secondary objectives were to describe the conversion and retention rates of this screening tool in a real-life setting.

## 2. Materials and Methods

Selection: We conducted an observational, retrospective, multicenter study. The Odysight^®^ application was proposed to patients with macular edema treated by intravitreal injection between 1 May 2019 and 30 June 2021 in three ophthalmological centers in Lyon, France (Croix-Rousse and Edouard Herriot University Hospitals, and the Desgenettes Military Hospital). Inclusion criteria were patients 18 years of age or older with macular edema of known etiology and needing recurrent intravitreal injection (exudative AMD, diabetic macular edema [DME], cystoid macular edema with retinal vein occlusion [RVO], myopic choroidal neovascularization [CNV], pachychoroid neovasculopathy, CNV due angioid streaks, or inflammatory CNV). Exclusion criteria were no smartphone or tablet, and patient refusal to use the application; patients unable to read or understand French were excluded, as were those with a binocular VA < 20/63 or a monocular VA < 20/200, for whom the application is not recommended by the editor (Tilak Healthcare). Informed consent was obtained from all the participants. Research was conducted in accordance with the Declaration of Helsinki. An international review board approved the study (Ethics Committee of the French Society of Ophthalmology, IRB 00008855 Société Française d’Ophtalmologie IRB#1). Odysight^®^ application: This application can be downloaded and used on a smartphone or tablet and requires an internet connection to be used. It is available in France and the USA on the Apple store^®^ and Google Play store^®^. A medical prescription is required to link the patient to his/her referring ophthalmologist. The application includes a VA test using the “tumbling E” and is based on the Early Treatment Diabetic Retinopathy Study (ETDRS) scale, which is the screening test and can lead to alerts. It also includes a puzzle game to reinforce adherence to the tests and an Amsler grid (not part of the alert system; Figure 1).

The first step in the application is a calibration test for each eye that defines baseline VA. For each VA test, the distance between the patient and the device is automatically measured by the camera of the smartphone or the tablet, as is the brightness. The tests are carried out in monocular fashion. Data are then sent to a digital platform to which only the prescriber has access. An alert can be triggered when a patient registers a drop of >5 VA letters on two consecutive tests compared to the threshold value, which is calculated according to the mean of previous tests (mean of a maximum of 15 previous tests). In the case of a drop of >5 VA letters compared to this threshold value, a pre-alert is triggered. The patient is then invited to redo a test to confirm this decrease of VA. If the decrease of >5 VA letters is confirmed, then the alert is triggered. The patient is then notified directly of this alert on the application but also by email and text message. It is explained to the patient to contact his/her ophthalmologist. The prescriber is also informed of the alerts during his/her connection to the secure platform and by an email. In addition, all patients were followed as usual, regardless of whether or not they used the application. Data collection: Patient characteristics collected were sex, age, disease, eye affected, best-corrected visual acuity (BCVA), intravitreal molecule (ranibizumab, aflibercept, dexamethasone implant), treatment regimen (reactive or proactive), and date of diagnosis. The VA was measured on the Snellen or ETDRS scales according to availability. All VA scores were converted to LogMAR for standardization and comparison. Data from OCT examinations were also collected. Recurrence of macular edema was defined as the presence of fluid on the OCT. A patient could have both eyes included in the study if both were treated with intravitreal injection. Outcome measures: The primary endpoint was the Sensitivity (Se) of alerts generated by the application; Se was the probability of an alert when the patient had macular edema recurrence. Specificity (Sp), positive predictive value (PPV) and negative predictive value (NPV) were also calculated: Sp was the probability of the absence of an alert when the patient did not have a recurrence, PPV was the probability of having a recurrence of macular edema if the test was positive and the NPV was the probability of not having a recurrence if the test was negative. These were calculated retrospectively by comparing the data from the application with that in the medical record completed during consultation including OCT. All patients who used the application (irrespective of the frequency) were considered, and a subgroup analysis was made where only patients who used the application on a regular basis (at least once a week) were analyzed. As an alert can only be triggered after a minimum of two tests on the same eye, only the patients that performed at least two tests on the studied eye were analyzed, and only the last test performed at home before the consultation was considered for comparison with the data collected during the consultation. If a recurrence was documented during the consultation, a treatment decision was indicated if the patient followed a reactive regimen, or the interval between two consultations/injections was reduced if the patient followed a proactive regimen. One patient could have several alerts. The secondary endpoints were the conversion rate (i.e., installation of the application and completion of at least one test in addition to the first calibration test), as well as the retention rate (i.e., use of the application more than 3 months after the first use) and the factors associated with this. The interval between the home test considered and the consultation was also calculated overall, and according to the agreement between the home test and the consultation. Statistics: Quantitative variables were described by the mean and a 95% confidence interval [95% CI] or by the median and interquartile range [IQR]; qualitative variables were described by frequency and percentage. Patient characteristics were compared between groups using a Chi-squared test or the Student’s *t*-test for qualitative parameters and the Wilcoxon test for quantitative parameters. A univariate analysis followed by multivariate logistic regression was conducted to look for predictive factors of retention. If both eyes were included in the study, the eye with the worst VA was taken into account for the multivariate analysis. The estimated duration of use was estimated using the Kaplan–Meier method. For the statistical analysis, three groups were composed according to the pathology and the expected number of patients: exudative AMD, DME or RVO, and other diseases. All analyses were performed using R-3.5.3 software (R Foundation for Statistical Computing, Vienna, Austria).

## 3. Results

### 3.1. Patients

A total of 149 eyes of 123 patients were included (Figure 2). The etiologies of macular edema were: exudative AMD (*n* = 69, 46.3%), RVO (*n* = 31, 20.8%), DME (*n* = 18, 12.1%), myopic CNV (*n* = 14, 9.4%), pachychoroid neovasculopathy (*n* = 12, 8.1%), post-operative macular edema (*n* = 2, 1.3%), CNV due to angioid streaks (*n* = 2, 1.3%), and inflammatory CNV (*n* = 1, 0.7%; Table 1).

### 3.2. Conversion and Retention Rate

A total of 64 patients (52.0%) were converted to the application (Figure 2). Among these, the median [IQR] duration of use was 96 [5–193] days. The retention rate in the total population at 3 months was 26.8% The estimated duration of use in the total population is estimated by a Kaplan–Meier curve (Figure 3). The median [IQR] number of tests among converted patients was 12 [2–31]; there were a median [IQR] 1.1 tests [0.6–2.5] performed per week per patient. Of the 46 patients (37.4%) who used the application more than 1 week, 21 (45.6%) performed at least one test per week. In addition to the tests, 54.2% of the total population played the games included in the application.

### 3.3. Alerts

A total of 47 patients performed at least two tests on the study eye in addition to calibration, and therefore could have an alert (index patients). Of the 206 tests (index tests) performed before a consultation and that could be used, there were 40 alerts from 18 patients, of which 39 could be interpreted. One alert could not be analyzed because the patient was lost to follow-up and did not wish to return for a consultation (Figure 2). Of the 39 alerts analyzed, 33 (84.6%) led to a consultation performed within 7 days. The mean (SD) interval between the alert and the next consultation was 8.9 (14.8) days; this was 3.3 (2.7) days for true positives and 11.4 (17.0) days for false positives, this difference was not statistically significative (*p* = 0.10); this was 7.9 (12.3) for true negatives and 18.9 (23.8) days for false negatives, this difference was statistically significative (*p* < 0.005). Of the 39 alerts there were 12 true positives and 27 false positive (Figure 2). The Se of alerts for the detection of recurrence was 30.8% (95% CI [17.6; 44.0]) and its Sp was 83.7% (95% CI [73.2; 94.3]). The PPV of alerts was 30.8% (95% CI [17.6; 44.0]) and the NPV was 83.7% (95% [73.2; 94.3]; Table 2); in those performing at least one test per week, the Se was greater (47.8%, 95% CI [28.6; 67.0]) and the Sp lower (78.0% 95% CI [69.1; 93.0]; Table 3).

Of the 12/39 alerts that detected a recurrence, eight (20.5% of all alerts) resulted in the scheduling or advancement of an intravitreal injection. In the other four eyes, the alert was triggered just before a planned consultation and management was not changed. Of the eight alerts that changed management, patients were on a reactive regimen in seven cases (87.5%) and dexamethasone implant injections were used in five eyes (62.5%).

### 3.4. Analysis of Patients Who Retained the Application for More Than 3 Months

Among the total population, a better baseline VA was found to be significantly associated with a better retention in univariate analysis (OR = 0.04, 95% CI [0.001; 0.53]; *p* = 0.03). This was confirmed in the multivariate analysis (OR = 0.05, 95% CI [0.002; 0.62]; *p* = 0.045). There was no significant association for age, sex, follow-up regimen, pathologies and time from diagnosis (Table 4).

## 4. Discussion

In the present study, the sensitivity of the Odysight^®^ application seems low. This can be explained by early recurrence being identified on the OCT by an increase of an edema, possibly without causing loss of VA. However, the application allows only the detection of a decrease in VA and therefore edema recurrence could be missed. This should be kept in mind when analyzing the results of this study because the decision to retreat a patient is made as soon as macular edema recurs, and this generally precedes VA decline [14]. Therefore, the application cannot be used as a means to replace regular consultation but could help to extend monitoring at home. Interestingly, when the application sent an alert this led to a modification of the management of the disease by bringing forward treatment in a fifth of cases. This is of importance, as it is now well known that numerous, or large fluctuations, of macular thickness are associated with VA loss over the course of the disease [15,16]. PPV was also low, representing a lot of false positives. Similarly, this can be due to the nature of the measurements, i.e., visual acuity; it has been shown that this measurement is not constant, especially in patients that have low VA [17], and that a change of five letters may be insufficient to detect a significant clinical change [18,19]. Odysight^®^ tried to limit it by triggering an alert after two measurements of a VA drop, and by trying to measure VA in reproductible conditions; the application evaluates the distance and ambient luminosity during each test.

It is of note that the Se and Sp herein were lower than that reported in a previous study evaluating the Odysight^®^ application (Se 99% and Sp 94%) [13]. However, the gold standard considered in this study was a decrease in clinically evaluated VA, which indicates that the Odysight^®^ application is able to accurately detect vision loss, whether or not this is due to edema recurrence (of note, among the 19 alerts, six needed intravitreal injection, and therefore the PPV is close to 30%, as was the case herein for exudative recurrence) [13]. To improve early detection of edema recurrence, other applications have been developed using hyperacuity systems. These hyperacuity tests are able to detect minimum angle of resolution differences of 3 to 10 s of arc, a smaller difference than that detected by VA [20]. However, applications using this system do not seem to have a PPV that is better than that found for Odysight^®^ in the present study; for instance, it is higher than that reported by Teo et al., who found an indication of intravitreal injection in only 5/33 alerts in a study evaluating the Alleye^®^ application [21]. Other studies do report higher PPV, but the 60% PPV found by Islam et al. for the indication of reinjection was greatly overestimated because only those who had both an alert on the Alleye^®^ application and who had experienced a visual loss during the first UK lockdown due to COVID-19 were seen in consultation (i.e., only 28 patients of the total of 78 patients with an analyzable alert) [22]. In addition, the 80% PPV reported by Faes et al. for the indication of reinjection following an alert triggered by Alleye^®^ is also overestimated, as the follow-up appointment was not brought forward in case of an alert, alerts with extreme variations in score were not considered, and only patients performing the test once or more times a week were included [23]. The latter is of note, as the sensitivity of the application herein was increased when patients used it at least once a week. In addition, it has been recently reported that patients using Alleye^®^ had a better chance to have a VA gain >5 letters compared to those who did not use it, demonstrating the potential benefit of the application [24]. Other applications such as myVisiontrack^®^ have not yet been evaluated for their diagnostic performance [10,25].

One of the main barriers to widespread prescription of a home screening test is patient adherence. In the present study, it was found that about half of the patients were converted to the application, i.e., they installed it on a mobile device and performed at least one test in addition to the calibration tests. However, only a little more than a quarter after 3 months continued to use it. Patient age was not found to be a predictive marker of application retention, which may be surprising, as a study using global data on Odysight^®^ found a better retention among those aged 65–80 years [26]. Patients who have a good baseline VA were more inclined to use the application for more than 3 months. The use of the application and maybe of a smartphone might be more difficult for patients with a poor VA, thus reducing the use of this application over time. No statistical difference concerning retention was found concerning the follow-up regimen, but it is of note that most alerts that led to a change in management involved patients on a reactive protocol and that had a dexamethose implant. The conversion and retention rates at 3 months herein were lower than those reported in another study conducted in France (respectively, 61.5% and 51%); this may be due to the difference in study population, as most patients had intravitreal injections capped at 3 months or no intravitreal injection at inclusion [13]. Similarly, another study reported by Teo et al. evaluating Alleye^®^ found greater retention but was conducted during the COVID-19 lockdown period (in Singapore), when patients were no longer seen in consultation and the application was thus the only means of monitoring visual function [21].

The present study has several limitations. First, the retrospective nature of the study did not allow us to collect some data, such as the total number of patients to whom the application was proposed. The possession of a smartphone or tablet is an obligation to use the application and many elderly patients do not have this type of equipment or are not comfortable in its use. In addition, some models of smartphones are not compatible with the use of Odysight^®^, because the camera on the device does not allow us to calculate the distance between patient and device, as well as the luminosity. This can partially explain the low conversion rate. It would have been interesting to know why some patients were not converted but we were not able to obtain this information, as the data were collected retrospectively. Similarly, we did not collect whether or not the patient was aware of the VA drop in the case of a true positive that changed the management of the disease. In this case, the patient could have gone to the ophthalmologist more quickly. However, even if they were aware, the application urges the patient to contact the ophthalmologist and the latter is alerted by email of this drop in VA, allowing a modification of management. Another limitation is that the patient performs the tests at home, the only way to verify the result was to have a consultation; to limit the interval, we therefore decided to evaluate only the last test conducted before consultation, but this interval may be sufficient to observe a change in the disease, thus biasing the evaluation of this test, particularly for the interpretation of false negative tests. Finally, the real-life nature of the study brought heterogeneity to the study population, as different macular diseases were included. However, this shows how the application would really be used in a retinal center in a population regularly treated by intravitreal injections.

## 5. Conclusions

In conclusion, the Se and the PPV of the Odysight^®^ application to detect recurrence of macular edema are low and do not correspond to those expected by a screening test. Nevertheless, it is not meant to replace regular consultations and its use in addition to the usual follow-up of the patient can detect, in certain cases, an early recurrence and thus allow an anticipated readjustment of the treatment. Its use, particularly if it is regular, in patients under a reactive regimen and with a good visual acuity seems to be the ideal indication for prescription and requires other more specific studies in the context of these pathologies.

## Figures and Tables

**Figure 1 jcm-11-05010-f001:**
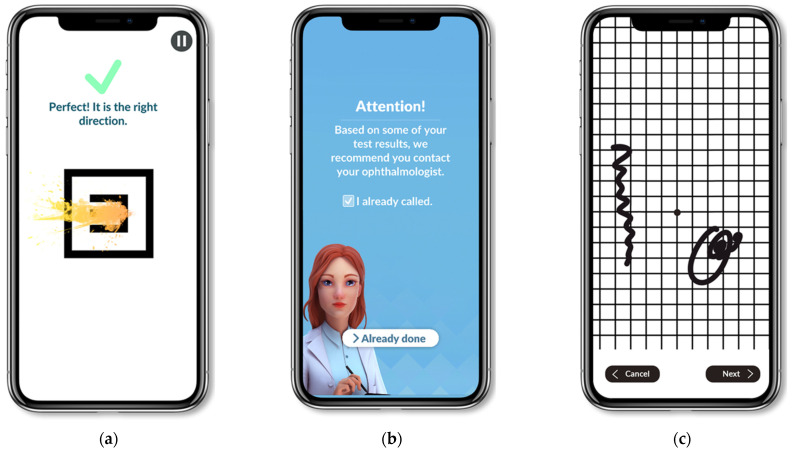
Images from the application: (**a**) VA test; (**b**) message when an alert occurs; (**c**) Amsler grid.

**Figure 2 jcm-11-05010-f002:**
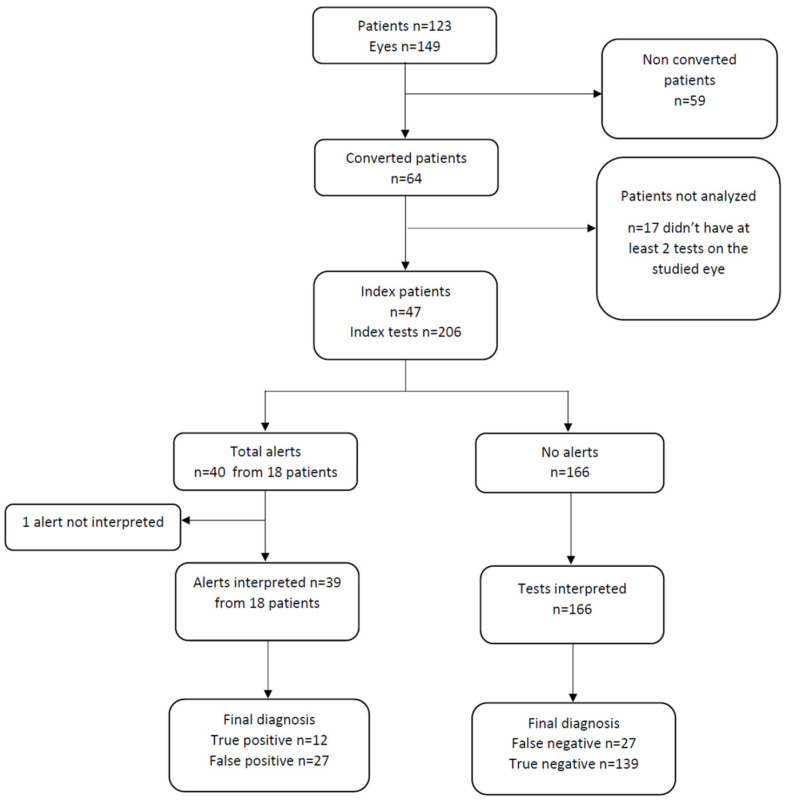
Flow chart.

**Figure 3 jcm-11-05010-f003:**
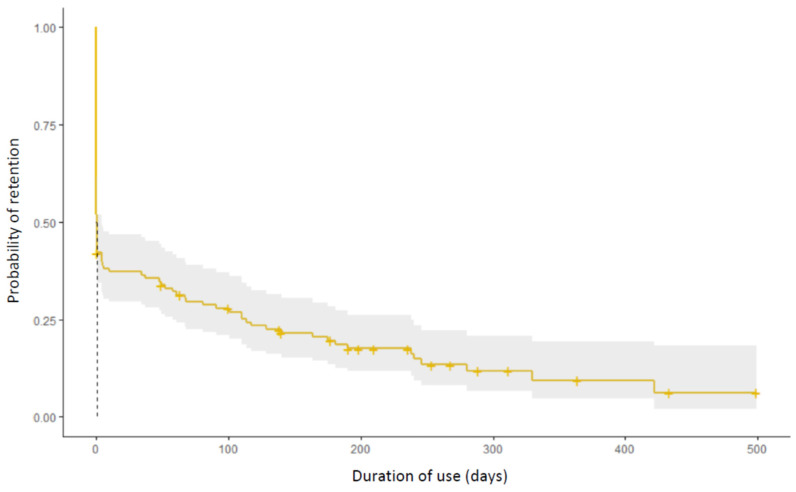
Estimated retention rate using the Kaplan–Meier method. Yellow line: probability of retention. Grey zone, 95% confidence interval, dotted line: median duration of use.

**Table 1 jcm-11-05010-t001:** Patient characteristics.

	*n* = 123
Sex, *n* (%)	
Female	72 (58.5)
Male	51 (41.5)
Age, years, *n* (%)	
<50	21 (17.1)
50–64	29 (23.6)
65–79	56 (45.5)
≤80	17 (13.8)
Disease, (*n* = 149) *n* (%)	
Exudative AMD	69 (46.3)
RVO–DME	31 (20.8)
Other	49 (32.9)
Follow-up, *n* (%)	
Reactive regimen	68 (55.3)
Proactive regimen	55 (44.7)
Interval between diagnosis and application prescription, *n* (%)	
<1 year	13 (10.6)
>1 year	110 (89.4)
Mean, years [95% CI]	3.0 [0.40; 16.1]
Baseline VA LogMAR, (*n* = 149)	
<0.2	96 (64.4)
0.2–0.4	34 (22.8)
≥0.5	19 (12.8)
Mean (SD)	0.2 (0.2)

AMD: age-related macular degeneration, DME: diabetic macular edema, RVO: retinal vein occlusion, other (post-operative macular edema, myopic choroidal neovascularization (CNV), pachychoroid neovasculopathy, inflammatory CNV, and CNV due to angioid streaks), VA: visual acuity, LogMAR: logarithm minimum angle of resolution.

**Table 2 jcm-11-05010-t002:** Characteristics of the Odysight^®^ test for exudative recurrence screening in total analyzable population (*n* = 47 patients).

Sensitivity [95% CI]	30.8 [17.6; 44.0]
Specificity [95% CI]	83.7 [73.2; 94.3]
Positive predictive value [95% CI]	30.8 [17.6; 44.0]
Negative predictive value [95% CI]	83.7 [73.2; 94.3]

**Table 3 jcm-11-05010-t003:** Characteristics of the Odysight^®^ test for exudative recurrence screening if patients used the application at least one time per week (*n* = 26).

Sensitivity [95% CI]	47.8 [28.6; 67.0]
Specificity [95% CI]	78.0 [62.1; 93.9]
Positive predictive value [95% CI]	35.5 [17.1; 53.9]
Negative predictive value [95% CI]	85.5 [72.0; 99.0]

**Table 4 jcm-11-05010-t004:** Factors associated with retention > 3 months.

	Retention > 3 Months		Analysis	
	No (*n* = 90)	Yes (*n* = 33)	Univariate OR [95% CI]	Multivariate OR [95% CI]
Sex, *n* (%)			*p* = 0.78	
Female	52 (57.7)	20 (60.6)	ref	ref
Male	38 (42.3)	13 (39.4)	OR = 0.89 [0.39; 1.99]	*p* = 0.73 OR = 0.86 [0.35; 2.07]
Age, years mean (min, max)	68.0 (23.0, 89.0)	69.0 (40.0, 87.0)	*p* = 0.59 OR = 1.01 [0.98; 1.04]	*p* = 0.65 OR = 1.01 [0.97; 1.05]
Disease, *n* (%)			*p* = 0.43	
Exudative AMD	41 (45.6)	16 (48.5)	ref	ref
RVO–DME	20 (22.2)	10 (30.3)	*p* = 0.61 OR = 1.28 [0.48; 3.31]	*p* = 0.86 OR = 1.10 [0.35; 3.48]
Other	29 (32.2)	7 (21.2)	*p* = 0.35 OR = 0.62 [0.21; 1.65]	*p* = 0.49 OR = 0.62 [0.15; 2.39]
Follow-up regimen *n* (%)			*p* = 0.47	
Reactive	48 (53.3)	20 (60.6)	ref	ref
Proactive	42 (46.7)	13 (39.4)	OR = 0.74 [0.32; 1.66]	*p* = 0.36 OR = 0.62 [0.23; 1.58]
Interval from diagnosis to application use *n* (%)			*p* = 0.74	
≤1 year	9 (10.0)	4 (12.1)	ref	ref
>1 year	81 (90.0)	29 (87.9)	OR = 0.81 [0.24; 3.2]	*p* = 0.95 OR = 0.95 [0.25; 4.22]
Visual acuity, LogMAR mean (SD)	0.2 (0.3)	0.1 (0.1)	***p* = 0.03** **OR = 0.04 [0.001; 0.53]**	***p* = 0.04** **OR = 0.05 [0.002; 0.62]**

OD: odds ratioAMD: age-related macular degeneration, DME: diabetic macular edema, RVO: retinal vein occlusion, other (post-operative macular edema, myopic choroidal neovascularization [CNV)], Pachychoroid neovasculopathy, inflammatory CNV and CNV due to angioid streaks), LogMAR: logarithm minimum angle of resolution.

## Data Availability

Not applicable.

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
