# Peer review of "Real-Life Value of the Odysight® Application in At-Home Screening for Exudative Recurrence of Macular Edema"

_jcm, 2022, doi:10.3390/jcm11175010_

Round 1

Reviewer 1 Report

Many terms are used that are not commonly found in the field of ophthalmology.

"Intravitreal injection" or "anti-vascular endotherial growth factor therapy" instead of "intravitreal therapy" at line 23.

"Cystoid macular edema with retinal vein occlusion" instead of "edematous retinal vein occlusion" at line 72.

"Myopic choroidal neovascularization" instead of "neovessels due to high myopia" at line 72.

"Pachychoroid neovasculopathy", which is involved in AMD, instead of "neovessels secondary to central serous chorioretinitis" at line 152.

CSC rarely involves edema.

The part referring to dexamethasone is weakly based and should not be included in the conclusion.

In how many of the 8 alerts that affected the treatment interval was the patient unaware? This is an important question because if the alert was issued after the patient was aware of drop of VA and then tested, it would have no clinical benefit.

To begin with, this study was not designed to evaluate Odysight, but only to describe what happened when Odysight was used. Therefore, unless the study is to compare the results with and without Odysight, the purpose of the study should be rewritten.

Reviewer 2 Report

In this manuscript, the authors studied the Real-life value of the Odysight® application in at-home screen for exudative recurrence of macular edema.

There are significant flaws in the literature review, design of the study, and discussion. The introduction section is based on old protocols of intravitreal injections. The information about the Amsler grid chart is not accurate. They have not discussed the peripheral hyperacuity meter. 

The study design is poor. They included patients with vision between 6/20 and 2/20 without justifying why they chose that range. They included patients with a wide range of diagnoses, making their results unreliable. For example, they had two patients with CSR. 

The discussion is contradictory. For example, " edema possibly leading to metamorphopsia and decreased contrast sensitivity or color vision, without necessarily causing loss of VA, whereas the application allows to detect only a decrease in VA; however, it is not meant to replace check-ups but to extend monitoring to the home." In addition, the discussion is not well organized and does not explain why the authors use Odysight and not peripheral hyperacuity and Amsler grid. 

The study does not support the conclusion.

Round 2

Reviewer 1 Report

In this revised version, the manuscript has been well improved according to the reviewers' comments.

Author Response

We thank the reviewer for his/her comment